# Determination of progressive stages of type 2 diabetes in a 45% high-fat diet-fed C57BL/6J mouse model is achieved by utilizing both fasting blood glucose levels and a 2-hour oral glucose tolerance test

Thuy Nguyen-Phuong[1,2,3], Sol Seo[1,2,3], Beum-Keun Cho[1,2,3], Jung-Ho Lee[1,2,3], Jiyun Jang[1,2,3], Chung-Gyu Park[1,2,3,4,5,6]*

1 Department of Microbiology and Immunology, Seoul National University College of Medicine, Seoul, South Korea, 2 Department of Biomedical Sciences, Seoul National University College of Medicine, Seoul, South Korea, 3 BK21Plus Biomedical Science Project, Seoul National University College of Medicine, Seoul, South Korea, 4 Cancer Research Institute, Seoul National University College of Medicine, Seoul, South Korea, 5 Institute of Endemic Diseases, Seoul National University College of Medicine, Seoul, South Korea, 6 Biomedical Research Institute, Seoul National University College of Medicine, Seoul, South Korea

* chgpark@snu.ac.kr, chgpark@gmail.com

**Data Availability Statement:** All relevant data are within the paper.

## Abstract

Type 2 diabetes is considered one of the top ten life-threatening diseases worldwide. Following economic growth, obesity and metabolic syndrome became the most common risk factor for type 2 diabetes. In this regard, high-fat diet-fed C57BL/6J mouse model is widely used for type 2 diabetes pathogenesis and novel therapeutics development. However, criteria for classifying type 2 diabetes progressive stages in this mouse model are yet to be determined, led to the difficulty in experimental end-point decision. In this study, we fed C57BL/6J male mice with 45% high-fat diet, which is physiologically close to human high-fat consumption, and evaluated the progression of type 2 diabetes. After consuming high-fat diet for 4 weeks, mice developed metabolic syndrome, including obesity, significant increase of fasting plasma cholesterol level, elevation of both C-peptide and fasting blood glucose levels. By combining both fasting blood glucose test and 2-hour-oral glucose tolerance test, our results illustrated clear progressive stages from metabolic syndrome into pre-diabetes before onset of type 2 diabetes in C57BL/6J mice given a 45% high-fat diet. Besides, among metabolic measurements, accumulating body weight gain > 16.23 g for 12 weeks could be utilized as a potential parameter to predict type 2 diabetes development in C57BL/6J mice. Thus, these results might support future investigations in term of selecting appropriate disease stage in high-fat diet-fed C57BL/6J mouse model for studying early prevention and treatment of type 2 diabetes.

**Funding:** This work was supported by the National Research Foundation of Korea (NRF) grant funded by the Korea government Ministry of Science and ICT (MSIT) (Grant No. 2019R1A2C2085287), and a grant from Seoul National University Hospital (2023). The funders had no role in study design, data collection and analysis, decision to publish, or preparation of the manuscript.

**Competing interests:** The authors have declared that no competing interests exist.

## Introduction

According to the International Diabetes Federation, the estimation of diabetes prevalence among people from 20 to 79 years old will be 783.2 million people globally by 2045, and medical costs associated with diabetes are projected to reach 1,054 billion USD [1]. It has become urgent to pay more attention and effort to prevent the disease, especially type 2 diabetes (T2D), as it accounts for over 90% of all diabetes cases.

Metabolic syndrome (MetS) with coexisting pre-diabetes symptoms, including obesity, dyslipidemia, hyperinsulinemia, and elevated fasting glucose, is a high risk factor for developing T2D. For instance, the incidence of MetS among T2D patients was 58.4% in Nepal, and 41.15% in South Korea [2, 3]. Additionally, subjects, who had MetS and pre-diabetes, were highly likely to have T2D with incident rate at about 53.1% and 47.6%, respectively. However, recovering from MetS and pre-diabetes decreased the T2D incident rate to only around 25% [3]. MetS and pre-diabetes prevalence worldwide are expected to be two to three times higher than T2D, so mechanism and treatment studies focusing on MetS and prediabetes stages might significantly contribute to T2D prevention, improve life quality and reduce economic burden globally [4].

C57BL/6J mouse strain naturally carries a missense mutation in the nicotinamide nucleotide transhydrogenase gene, resulting in weak insulin secretion by pancreatic β-cells under the high-dose glucose stimulation [5, 6]. Due to this feature, C57BL/6J mice are prone to develop T2D by feeding with various diets containing high calories from fat compared to other mouse strains and thus has become a preference of choice for a high-fat diet (HFD)-induced T2D mouse model [7, 8]. Although HFD-fed C57BL/6J mice quickly developed MetS, the pre-diabetes stage is yet to be established, leading to either difficulty determining the study end-point or controversial reports among research groups [9]. For instance, C57BL/6J mice subjected to high-fat, high sucrose condition were considered to have T2D by 12 weeks, whereas the other group fed mice with a similar diet for 22 weeks to achieve pre-diabetes [10, 11].

Our study aimed to identify MetS and pre-diabetes simultaneous stage in C57BL/6J mice subjected to 45% HFD by utilizing both fasting blood glucose level (F-BGL) and the 2-hour-oral glucose tolerance test (2-hour-OGTT), the gold standard applied in humans. Our data illustrated that C57BL/6J mice displayed pre-diabetes phenotype as early as eight weeks on HFD, while the onset of T2D appeared after feeding HFD for 16 weeks. Additionally, body weight might be a valuable parameter for monitoring T2D progression in HFD-fed C57BL/6J mice, providing benefits for future investigation.

## Materials and methods

### Mice

Male C57BL/6J mice were obtained from Jackson Laboratory (Bar Harbor, ME) at four weeks of age and maintained in the Institution for Experimental Animals at Seoul National University College of Medicine. The mice were housed in a pathogen-free barrier facility and kept at a 12-hour light/dark cycle. After two weeks of stabilization, mice were divided randomly into two groups (four mice per cage) and fed with a regular diet (labeled as ND group) or a high-fat diet (labeled as HFD group) for 20 weeks. The regular diet contained 12.41 Kcal% of fat, 24.52 Kcal% of protein, and 63.07 Kcal% of carbohydrates (2.97 Kcal/g; 38057; Purina Irradiated Lab, Seongnam, Korea). A high-fat diet contained 45 Kcal% of fat, 20.0 Kcal% of protein, and 35.0 Kcal% of carbohydrates (4.7 Kcal/g; D12451; Research Diet, Inc., New Brunswick, NJ). A total of 64 mice were divivied as 4 mice per cage, 8 cages per group. 1 mouse from ND group was excluded due to weight loss and was excluded from all analysis. Body weight and fasting

blood glucose level were measured biweekly. Blood glucose levels were tested after six-hour fasting by One Touch Ultra glucometer (Lifescan, Inc., Chesterbrook, PA, USA).

All experiments were conducted in accordance with the National Institutes of Health (NIH) Guide for the Care and Use of Laboratory Animals and under the approval of the Institutional Animal Care and Use Committee (IACUC) of SNU (Approval number SNU-201103-3-4). All animal experiments followed ARRIVE guidelines.

### Oral glucose tolerance test

Mice were fasted for 6 hours from 8 am to 2 pm, and then 1 mg of glucose (20% dextrose) per gram body weight was fed through oral gavage. Blood glucose levels were measured before, at 5, 10, 15, 30, 60, 90, and 120 minutes after the gavage using One Touch Ultra glucometer (Lifescan, Inc., Chesterbrook, PA, USA) by snipping off the end of the tail.

### Biochemistry test

At indicated weeks, mice were fasted for 6 hours before collecting blood via retro-orbital bleeding under anesthesia using a heparinized capillary tube (Marienfeld Superior, Lauda Königshofen, Germany). Blood was transferred to Eppendorf tubes containing 0.1% v/v Ethylenediaminetetraacetic acid. Plasma was collected by centrifugation at 2000 RPM for 20 minutes at room temperature. All plasma samples were stored at minus 20 degrees Celsius until analyzed. Fasting plasma C-peptide was measured by Mouse C-Peptide ELISA kit (90050; Crystal Chem, Brook Drive, IL). All procedures were performed according to the manufacturer's instructions.

Fasting plasma cholesterol and triglyceride levels were measured with HITACHI 7180 (Hitachi High-Tech Korea Co, Ltd, Seongnam, KOREA). All procedures were performed according to the manufacturer's instructions.

### Data analysis

All statistical analyses were performed using GraphPad Prism version 9.5.1 (GraphPad Software, San Diego, CA, USA). Data were expressed as mean ± standard error of the mean (SEM). The comparisons between HFD and ND groups on the change over time of body weight, cholesterol, triglyceride, C-peptide, energy and water intake, F-BGL, 2-hour-OGTT were assessed using two-way analysis of variance (ANOVA), followed by Holm-Šídák multiple comparisons test for each time point. For oral glucose tolerance test, area under the curve (AUC) was calculated by trapezoid rule, and the comparison was done by two-tailed student's t-test.

## Results

### Metabolic syndrome development in 45% HFD-fed C57BL/6J mice

Over a twenty-week period, mice gradually gained weight regardless of the diet type (**Fig 1A**). As expected, mice fed with HFD became significantly heavier than those fed with ND as early as the second week ($p$ = 0.02) (**Fig 1A**). When comparing the relative weight gain to their initial body weight at six weeks of age, the HFD group showed a more rapid increase rate than the ND group (**Table 1**). The mean value of the relative weight gain in the HFD group reached to the mean value + 3 standard deviations of the ND group by the fourth week (**S1 Fig**). This result suggested that C57BL/6J male mice became obesity after four weeks of consuming a 45% HFD. The fasting plasma cholesterol of the HFD group was significantly higher than that of the ND group from the fourth week and kept increasing up to twenty weeks (**Fig 1B**). Meanwhile, the fasting plasma triglyceride level in the HFD group was lower than in the ND group

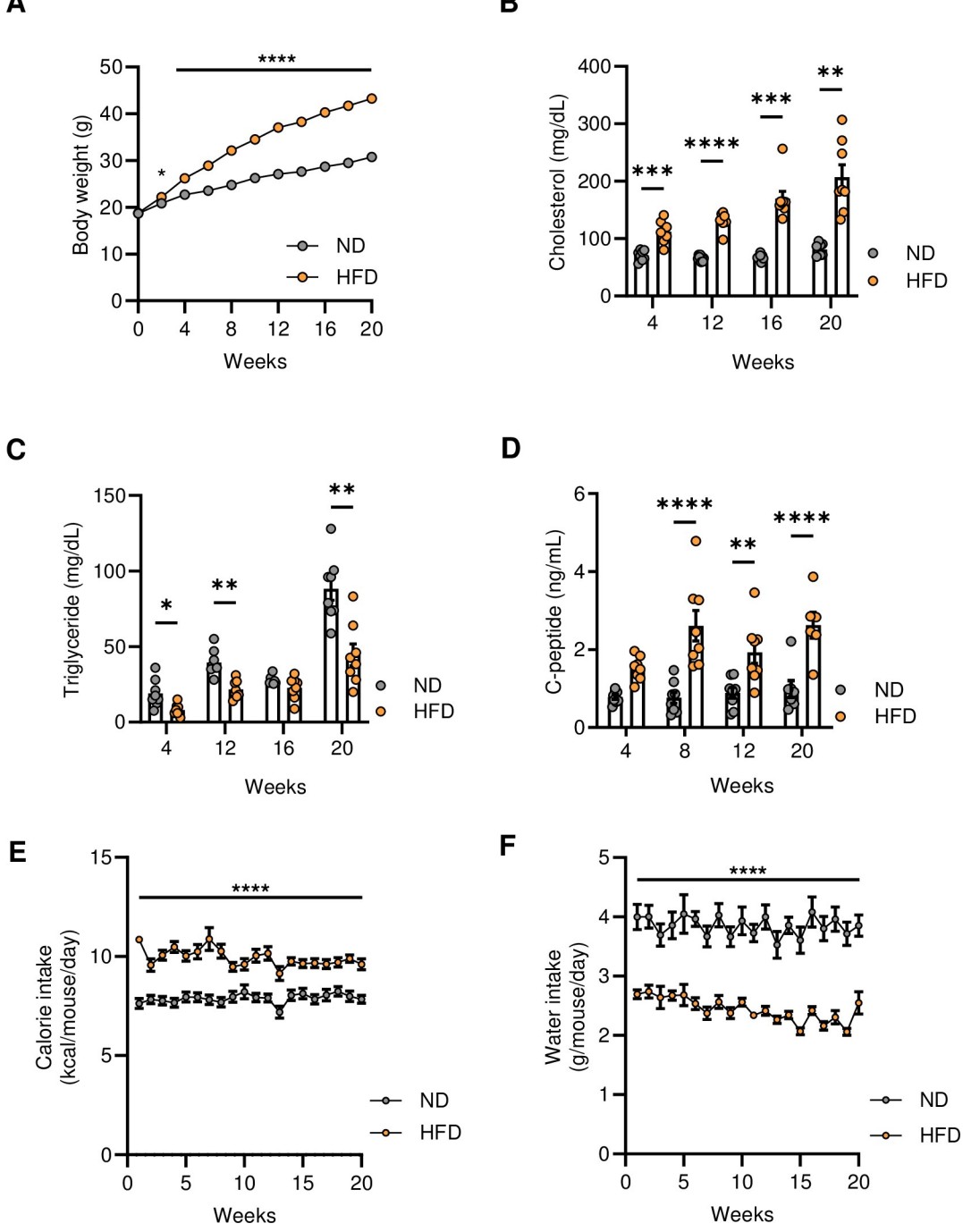

**Fig 1. Metabolic syndrome features of C57BL/6J mice fed 45% HFD. (A)** Body weight gain of mice fed with ND (n = 31) or 45% HFD (n = 32). Changes in six-hour fasting plasma (**B**) cholesterol, (**C**) triglyceride, and (**D**) C-peptide during 20 weeks (n = 6–8 mice per group). (**E**) Calorie intake, and (**F**) water intake were measured weekly. Data were analyzed using two-way ANOVA with Holm-Šídák multiple comparisons test and represented as mean ± SEM, ns not significant, * p < 0.05, ** p < 0.01, *** p < 0.001, **** p < 0.0001.

(**Fig 1C**). Next, we examined the endogenous insulin production by measuring the fasting plasma C-peptide. Although we observed an increase in C-peptide concentration after four weeks on HFD, this metabolic parameter showed a significantly higher level in the HFD group

**Table 1. Relative weight gain to the initial body weight during 20 weeks.**

| Weeks | ND (n = 31) mean ± SEM | HFD (n = 32) mean ± SEM |
|---|---|---|
| 0 | 1 ± 0 | 1 ± 0 |
| 2 | 1.12 ± 0.04 | 1.18 ± 0.06 |
| 4 | 1.22 ± 0.06 | 1.4 ± 0.12 |
| 6 | 1.26 ± 0.04 | 1.54 ± 0.14 |
| 8 | 1.33 ± 0.05 | 1.71 ± 0.15 |
| 10 | 1.4 ± 0.09 | 1.84 ± 0.17 |
| 12 | 1.44 ± 0.1 | 1.97 ± 0.18 |
| 14 | 1.47 ± 0.09 | 2.04 ± 0.19 |
| 16 | 1.52 ± 0.12 | 2.15 ± 0.19 |
| 18 | 1.57 ± 0.13 | 2.22 ± 0.18 |
| 20 | 1.64 ± 0.13 | 2.31 ± 0.18 |

The increase in weight gain over the initial body weight at six weeks of age was calculated

than in the ND group after eight weeks (**Fig 1D**). The mean value of F-BGL of HFD group also elevated higher than that of the ND group at the fourth week, yet the level still sustained below 200 mg/dL (**S1 Table**). Weekly food and water monitoring showed more energy consumption rate and less water intake in the HFD group (**Fig 1E and 1F**). These data confirmed that C57BL/6J mice developed metabolic syndrome after consuming 45% HFD for four weeks and kept worsening during examination period.

## Deterioration of glycemic control during 20 weeks consuming 45% HFD

Next, to detect the effect on glucose metabolism induced by HFD in C57BL/6J mice, F-BGL was tested biweekly. The HFD group has significantly increased F-BGL from the fourth week compared to the ND group, yet the level was still about 197.3 mg/dL ± 3.8 (n = 32), which is within the normal range of this mouse strain. After six weeks of feeding HFD, F-BGL crossed the 200 mg/dL threshold and sustained a steady high level of around 210 mg/dL from the eighth week onward (**Fig 2A** and **S1 Table**). Mice also underwent an oral glucose tolerance test (OGTT) every four-weeks (4, 8, 12, 16, and 20 weeks) to evaluate the peripheral glucose clearance capability. The level of blood glucose was measured at 0, 5, 10, 15, 30, 60, 90, and 120 minutes via tail vein. The area under the curve of the OGTT was significantly higher in the HFD group than in the ND group as early as the fourth week (**Fig 2B–2F** and **S2 Fig**). Besides, we also interpreted these data as 2-hour-OGTT. In agreement with above analysis, 2-hour-OGTT was significant higher in HFD group than in ND group. Also, comparison within the HFD group showed that 2-hour-OGTT at week 16 and 20 were significant higher than that at week 4 (**Fig 2G**). Interestingly, 2-hour-OGTT illustrated a worsening trend of glucose clearance in the HFD group with three distinct phases: (1) 222.2 ± 13.1 mg/dL after the first four weeks, (2) from 265.3 ± 18.6 to 309.8 ± 17.7 mg/dL during 8 to 12 weeks, and (3) maintained around 378.1 ± 38.9 to 378.6 ± 43 from 16 to 20 weeks (**Fig 2G**). In contrast, the age-matched control group on ND displayed consistent results at around 180 mg/dL (**Fig 2G**). Together, these data exhibited the decline in glycemic control induced by 45% HFD consumption with clear-cut stages.

## Body weight gain in C57BL/6J mice subjected to 45% HFD mimicks human's risk factor for T2D development

Our above data demonstrated the worsening of metabolic parameters coincident with failure from mild to moderate glycemic control during a period of 20 weeks in C57BL/6J male mice

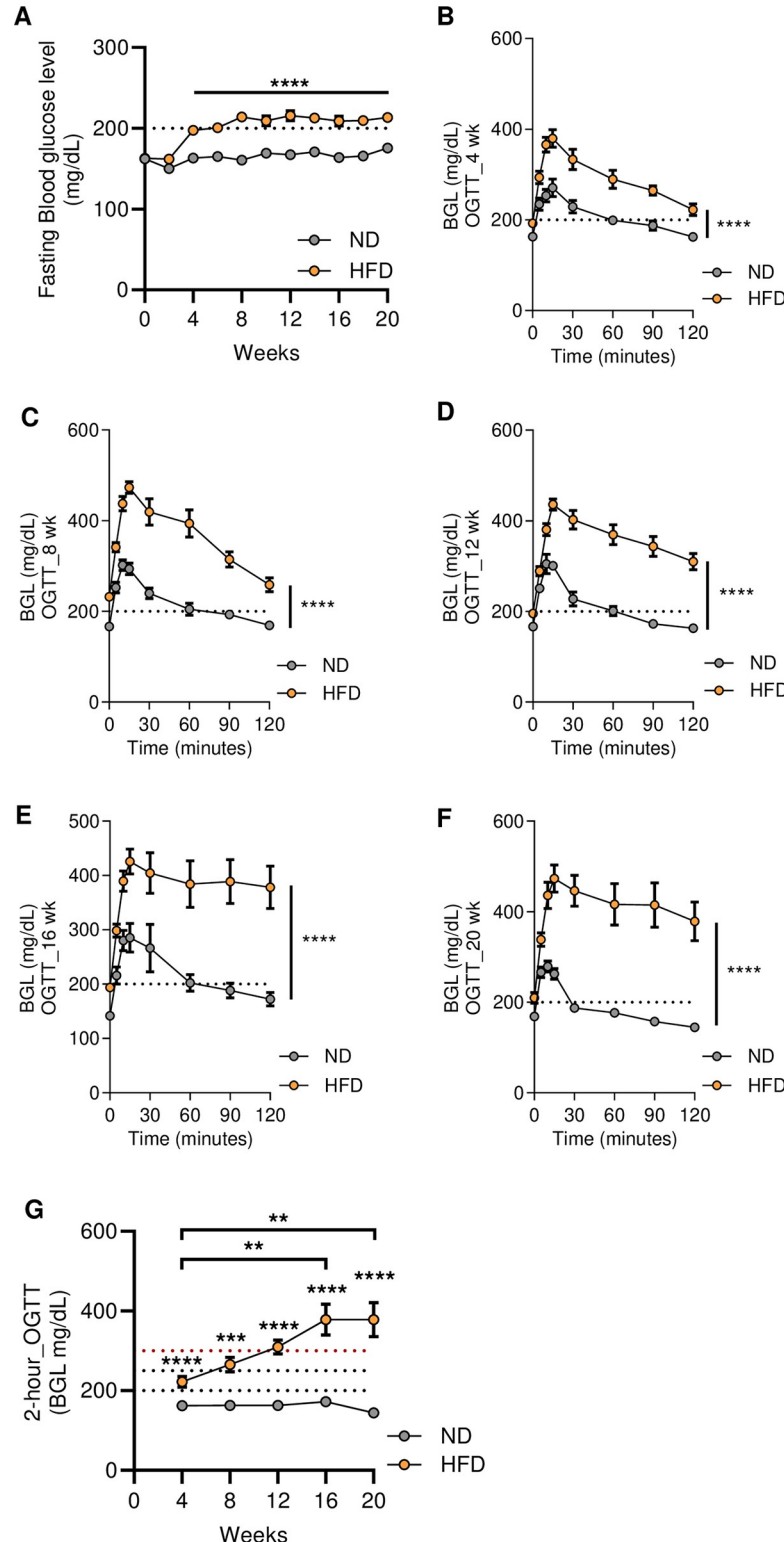

**Fig 2. Monitoring F-BGL and OGTT revealed the progression of T2D from MetS in HFD-fed C57BL/6J mice.** (**A**) F-BGL at two-week intervals before and after feeding with 45% HFD (n = 31 for ND group, n = 32 for HFD group). OGTT was examined at (**B**) 4 weeks, (**C**) 8 weeks, (**D**) 12 weeks, (**E**) 16 weeks, (**F**) 20 weeks, and the area under the curve was calculated to compare ND and HFD groups. (**G**) Blood glucose was measured after two hours of OGTT (**B–G**, n = 8 mice per group). OGTT results were represented for two to three independent experiments. Data represent

mean ± SEM. *** p < 0.001, **** p < 0.0001 by 2-way ANOVA with Holm-Šídák multiple comparisons test, and two-tailed student's t-test.

fed with 45% HFD. We questioned which metabolic factor tested above, including weight gain, cholesterol, and triglyceride could predict the loss of glycemic control in this HFD mouse model. First, the association of those factors with F-BGL during 20 weeks was examined using mice with a full diagnostic panel (weight gain, cholesterol, triglyceride, and F-BGL) at week 4 (n = 16), week 12 (n = 29), week 16 (n = 16), and week 20 (n = 16). A mouse from each time point was considered as an independent subject, thus a total of n = 77 subjects was input for multi-variables analysis. Matrix correlation result showed a positive correlation among body weight, fasting plasma cholesterol and F-BGL (**Fig 3A**). This data confirmed that HFD-fed C57BL/6J male mouse model reflexs the human disease, in which individuals having MetS and/ or pre-diabetes are at high risk of T2D development.

Next, because weight gain had a stronger positive correlation with F-BGL compared to fasting plasma cholesterol, we identified which amount of body weight gain could predict the decline in glycemic control of C57BL/6J mice. Since mice developed mild glycemic control failure from week 8 with 2h-OGTT over 250 mg/dL (**Fig 2G**), at week 8, 12, 16 and 20, mice were grouped depending on their F-BGL either less than 200 mg/dL (as normal glucose level) or equal to and more than 200 mg/dL (abnormal glucose level) for receiver operating characteristic curve (ROC) analysis. Area under the ROC curve revealed that weight gain at week 12 is likely to predict a mouse's failure to control its F-BGL (AUC = 0.9373), with suggestion of "cut-off" for weight gain > 16.23 g (85.71% of sensitivity and 90.24% of specificity, likelihood ratio 8.786) (**Fig 3B–3E**). To validate this "cut-off" value, we re-analyzed the 2-hour-OGTT in the HFD group at week 16 and 20. Interestingly, mice gained less than 16.23 g at week 12 had 2-hour-OGTT at week 16 and 20 below 300 mg/dL (**Fig 3F**). These data suggested that although mice fed with 45% HFD developed MetS together with impairment of glycemic control, those that gained weight over 16.23 g after 12 weeks were likely to develop hyperglycemia.

## Discussion

This study has classified the MetS and pre-diabetes stages before turning into T2D in the 45% HFD-fed C57BL/6J mice. Specifically, mice that consumed 45% HFD for four weeks appeared to have metabolic disorders, including obesity, dyslipidemia, and non-significant increase of fasting plasma C-peptide. Meanwhile, the elevation of F-BGL was still less than 200 mg/dL, the basal level of C57BL/6J strain [7]. These results suggested that C57BL/6J mice subjected to 45% HFD for four weeks could be diagnosed as MetS, and the increased endogenous insulin production was just a normal adaptation to an acute nutrient overloading rather than pre-diabetes or T2D.

In humans, the gold standard for distinguishing pre-diabetes from T2D is F-BGL or 2-hour OGTT, and omitting OGTT might lead to misdiagnosis of pre-diabetes [12, 13]. According to the American Diabetes Association, a pre-diabetes individual is defined as having an intermediate level of F-BGL and/ or 2-hour-OGTT [12]. In rodents, several parameters were used to define prediabetes, including glucose intolerance, hyperinsulinemia, hyperlipidemia, mild hyperglycemia and increased 2-hour glucose tolerant test, yet the detailed criteria for diagnosis is still lacking [9]. In this study, 45% HFD fed C57BL/6J mice from 8 to 20 weeks showed two-stage elevation of 2-hour-OGTT with cut-point of approximately 300 mg/dL. Thus, our data proposes that 45% HFD induced pre-diabetes from the eighth week with 2-hour-OGTT range from 250 to 300 mg/dL before turning to the onset of T2D from the sixteenth week with 2-hour-OGTT over 300 mg/dL in C57BL/6J mice. Indeed, 2-hour OGTT in mice was

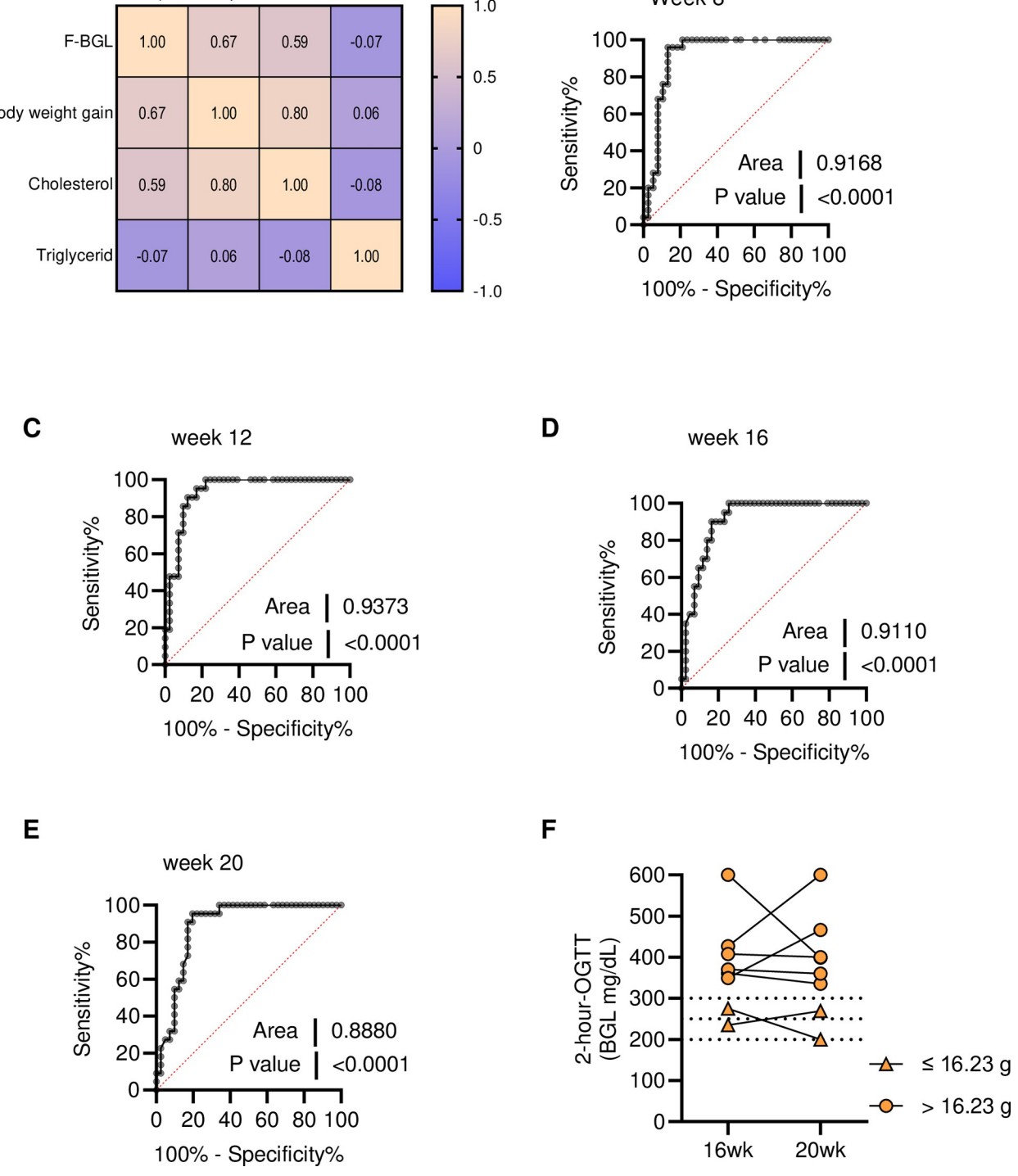

**Fig 3. Body weight gain could be utilized to estimate the development of T2D in C57BL/6J mice.** F-BGL and metabolic parameters, including weight gain, cholesterol, and triglyceride from four to twenty weeks, were used; each time point was considered as an independent subject, a total of n = 77 were pooled for multi-variables analysis, and data presented as (**A**) correlation matrix. Sensitivity and specificity of weight gain at (**B**) 8 weeks, (**C**) 12 weeks, (**D**) 16 weeks, and (**E**) 20 weeks predict T2D in C57BL/6J mice were analyzed by ROC (n = 63 at each time point). (**F**) Validation of the weight gain "cut-off" value by 2-hour-OGTT of HFD group (n = 8).

suggested to reach at least 300 mg/dL to be equivalent to the T2D diagnostic criteria of 200 mg/dL in human that might be due to the higher level of basal blood glucose level in mice [14]. Of note, mice are nocturnal feeders with a majority of food intake occurring during the dark cycle, and overnight fasting was shown to enhance insulin sensitivity and glucose uptake by muscle [15]. Additionally, glucose administration through oral gavage is absorbed by gastrointestinal tract, leading to gut incretin hormones-stimulated insulin secretion [16]. In this study, the glucose tolerance test was conducted after a six-hour fasting from 8 am to 2 pm through oral gavage to resemble human physiological conditions, the range of the 2-hour glucose tolerance test obtained either from a longer fasting duration or by intraperitoneal injection method might need to be evaluated separately.

Besides, the 2-hour-OGTT from 4 to 12 weeks increased gradually with no significant difference among time points in the HFD group might illustrate the accumulating effect of chronically consuming a diet with high calories from fat. It pointed out the importance of feeding duration in the HFD-induced T2D in the C57BL/6J mouse model. In addition, our data suggested accumulating weight gain over 16.23 g after 12 weeks fed with 45% HFD as the possible predictor of T2D development in C57BL/6J mice. Therefore, focusing on the mice likely to develop T2D from MetS and pre-diabetes might expand our knowledge on the disease pathogenesis. The limitation of this model is the lower fasting plasma triglyceride level in the HFD group compared to the normal diet group, which might be due to the triglyceride storage in the intestine of mice fed with 45% HFD [17, 18].

In summary, our study illustrated an overview of the early progressive stage from MetS and pre-diabetes to onset T2D in C57BL/6J male mice by feeding with 45% HFD, which might support future investigations in the field of T2D prevention and treatment at the early stage of the disease.

## Supporting information

**S1 Fig. The mean of relative weight gain with 3 standard deviations over 20 weeks.** Mean relative weight gain of ND group (black, solid line) with their upper 3 standard deviations (black, dash line). Mean relative weight gain of HFD group (orange, solid line) with their lower 3 standard deviations (orange, dash line).
(TIF)

**S2 Fig. Area under the curve analysis of the OGTT test over 20 weeks.** OGTT test were compared between ND group and HFD group at (A) week four, (B) week eight, (C) week twelve, (D) week sixteen, and (E) week twenty by area under the curve analysis. Data represent mean ± SEM. *** $p < 0.001$, **** $p < 0.0001$ by two-tailed student's t-test.
(TIF)

**S1 Table. F-BGL changes during 20 weeks.** Mean ± SEM value of F-BGL in both HFD and ND groups. p-value was calculated by 2-way ANOVA with Holm-Šídák multiple comparisons test.
(TIF)

**S1 File. Raw data of body weight, cholesterol, triglyceride, C-peptide, calorie and water intake, F-BGL, and 2h-OGTT, including F-values and degrees of freedom.**
(PDF)

## Author Contributions

**Conceptualization:** Chung-Gyu Park.

**Data curation:** Thuy Nguyen-Phuong, Sol Seo, Beum-Keun Cho, Jung-Ho Lee, Jiyun Jang.

**Formal analysis:** Thuy Nguyen-Phuong.

**Methodology:** Thuy Nguyen-Phuong, Chung-Gyu Park.

**Supervision:** Chung-Gyu Park.

**Validation:** Thuy Nguyen-Phuong.

**Visualization:** Thuy Nguyen-Phuong.

**Writing – original draft:** Thuy Nguyen-Phuong.

**Writing – review & editing:** Thuy Nguyen-Phuong, Chung-Gyu Park.

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
