## [Decision Letter · Decision Letter 0]

29 Aug 2023

PONE-D-23-24628Determination of progressive stages of type 2 diabetes in a 45% high-fat diet-fed C57BL/6J mouse model is achieved by utilizing both fasting blood glucose levels and a 2-hour oral glucose tolerance testPLOS ONE

Dear Dr. Park,

Thank you for submitting your manuscript to PLOS ONE. After careful consideration, we feel that it has merit but does not fully meet PLOS ONE’s publication criteria as it currently stands. Therefore, we invite you to submit a revised version of the manuscript that addresses the points raised during the review process.

We look forward to receiving your revised manuscript.

Kind regards,

Peng Gao, Ph.D.

Academic Editor

PLOS ONE

Journal Requirements:

"This work was supported by the National Research Foundation of Korea (NRF) grant funded by the Korea government Ministry of Science and ICT (MSIT) (Grant No. 2019R1A2C2085287), and a grant from Seoul National University Hospital (2023)."

"This work was supported by the National Research Foundation of Korea (NRF) grant funded by the Korea government Ministry of Science and ICT (MSIT) (Grant No. 2019R1A2C2085287), and a grant from Seoul National University Hospital (2023). The funders had no role in study design, data collection and analysis, decision to publish, or preparation of the manuscript."

Reviewers' comments:

Reviewer's Responses to Questions

**Comments to the Author**

1. Is the manuscript technically sound, and do the data support the conclusions?

Reviewer #1: Yes

Reviewer #2: Partly

2. Has the statistical analysis been performed appropriately and rigorously? 

Reviewer #1: Yes

Reviewer #2: I Don't Know

3. Have the authors made all data underlying the findings in their manuscript fully available?

Reviewer #1: Yes

Reviewer #2: No

4. Is the manuscript presented in an intelligible fashion and written in standard English?

Reviewer #1: Yes

Reviewer #2: Yes

5. Review Comments to the Author

Reviewer #1: This manuscript explored the use of fasting glucose and glucose tolerance test for diabetes diagnosis in mice. Here are some questions:

1. The abstract needs more specific results, e.g., by which week MetS develops and based on what parameters, body weight increase by what percentage can be used as a predictor of hyperglycemia.

2. Results. Line 117-118. By which week the weight gain was different between high-fat and ND group? This is probably more important than comparison to the baseline given that there is healthy weight gain even when the mice are on an ND.

3. Line 126. Was there any difference in calorie intake?

4. Line 127. What are the criteria for MetS diagnosis? It seems that at week 4 only weight and cholesterol were significantly different between HF and ND, doesn’t MetS diagnosis also need a blood pressure or blood glucose measurement? A relevant marker C-peptide was not different until week 8.

Some typos need to be corrected

Line 24. “are” is missing after “mouse model”; line 42. “of” should be deleted

Reviewer #2: This study aims to establish criteria with which to distinguish progressive stages in the development of diet-induced Type 2 diabetes (T2D) in mouse studies, with which investigators might more accurately focus studies on prevention and treatment. They do this by examining fasting blood glucose levels, oral glucose tolerance test performance, and body weight gain in male C57BL/6J mice, a popular model for studying diet-induced obesity, over a course of 20 weeks on high-fat diet. The results are potentially useful in the study of the development of T2D should these stages prove to be as distinct as purported. However, the manuscript as written has several deficiencies that should be resolved in order to be thoroughly convincing.

Major

1. The main goal of the study is to distinguish metabolic syndrome from pre-diabetes and eventually from onset of type 2 diabetes. The authors appear to propose that metabolic syndrome (characterized by obesity, dyslipidemia, and slight increase in fasting plasma C-peptide) differs from pre-diabetes by the lack of both elevated fasting blood glucose level (F-BGL) or significant elevation of 2-hour oral glucose tolerance (OGTT) values above “normal” levels, while pre-diabetes is characterized by elevated F-BGL intermediate but significantly elevated 2-hour OGTT, and diabetes with elevated F-BGL and even higher OGTT levels. However, the purported purpose of this study is to propose criteria for classifying type 2 diabetes progressive stages, yet operational definitions of these stages are never defined. For example, at what point in another group’s study would one consider the mice pre-diabetic versus diabetic?) It would be helpful to explicitly and clearly state these criteria if indeed this is the main assertion.

2. This study only addresses development of diabetes in male mice in this model, which differs in several relevant regards from female responses in several of the measures analyzed (e.g., weight gain in females is much less dramatic during development of diabetes). Conclusions of this study should be clearly explicitly confined to male mice.

3. Please clarify which factors were included in the two-way ANOVAs employed in the analyses, as well as associated F-values, degrees of freedom and “n” for the tests in question.

4. In the description of “Deterioration of glycemic control during 20 weeks consuming 45% HFD”, authors refer to “area under the curve” of the OGTT (line 139), yet the data presented in the accompanying Fig. 2 are not AUC data (but would be a relevant analysis). Moreover, if AUC data were analyzed, the method of calculating AUC should be described in the Methods section.

5. The authors’ make little mention of the methods of the correlational multi-variable analysis, the results of which suddenly appear in the manuscript (line 156). Similarly, there is no description of the receiver operating characteristic curve (ROC) analysis (line 163). These descriptions are essential for appropriate evaluation of these results.

6. The “cut-offs” for body weight gain at which body weight predicted loss of glycemic control (line 166) are stated in absolute values (e.g., 37.6g). Given that the weights at which the stages of diabetes development were unique to this specific cohort of mice given HFD at a specific age, would it not be more useful to other groups to determine relative weight gain cut-offs (i.e., relative to starting weights? Or weight gain relative to that in control mice?) that would be more extrapolatable to studies in which there is some variability from this study, such as in the age at which HFD is begun, in the starting weights of the mice, the rate of growth in the different groups in the study, or other factors? (For example, if one were to perform a study where mice received HFD at 5 weeks of age, would 37.6g be the body weight at which this tipping point occurs?)

7. It is unclear from Ref. 14 how “six-hour fasting from 8 am – 2 pm” resembles human physiological conditions (line 190). Is it the duration of the fast, or the timing of the fast that is deemed relevant here? Does the fact that mice are nocturnal and normally sleeping (not eating) at 8 am not put this method at odds with human conditions of the OGTT?

8. The authors claim that all data is contained within the manuscript, yet only summary statistics are included.

Minor issues:

9. Fasting blood glucose levels were not “tested… followed by six-hour fasting.” (line 83) Presumably the levels were tested following six-hour fasting?

10. It is unclear what the authors mean by “cutting off the tail vein” (line 92). Does this mean snipping off the end of the tail, rather than via a puncture method? Please define this more clearly.

11. Authors claim that “mice fed with HFD gradually gained weight” (line 116). From the data, it appears that both groups of mice gained weight, although HFD mice gained more.

6. PLOS authors have the option to publish the peer review history of their article (what does this mean?). If published, this will include your full peer review and any attached files.

Reviewer #1: No

Reviewer #2: **Yes: **E. Todd Weber

---

## [Author Response · Author response to Decision Letter 0]

5 Oct 2023

October 5th, 2023

Dear Dr. Peng Gao

PLOS ONE Academic Editor,

Thank you for these comments designed to improve our manuscript, "Determination of progressive stages of type 2 diabetes in a 45% high-fat diet-fed C57BL/6J mouse model is achieved by utilizing both fasting blood glucose levels and a 2-hour oral glucose tolerance test " [PONE-D-23-24628]. We greatly appreciate the time and effort that reviewers and editors have dedicated to providing valuable feedback on our work. We have tried our best to address all the comments from reviewers. The changes were included within the revised manuscript (labeled as “Revised manuscript with track changes”). Here is a point-by-point response to the reviewers’ comments and concerns.

Journal Requirements:

-> Thank you for the comment. We have prepared the revised manuscript according to the style template. 

"This work was supported by the National Research Foundation of Korea (NRF) grant funded by the Korea government Ministry of Science and ICT (MSIT) (Grant No. 2019R1A2C2085287), and a grant from Seoul National University Hospital (2023)."

"This work was supported by the National Research Foundation of Korea (NRF) grant funded by the Korea government Ministry of Science and ICT (MSIT) (Grant No. 2019R1A2C2085287), and a grant from Seoul National University Hospital (2023). The funders had no role in study design, data collection and analysis, decision to publish, or preparation of the manuscript."

-> Thank you for the comment. We have removed all funding-related text from the revised manuscript. We would like to keep our Funding Statement as mentioned in the cover letter.

-> Thank you for the comment. We have successfully updated ORCID ID (0000-0003-4083-8791) for the corresponding author in Editorial Manager.

Reviewers' comments:

Reviewer's Responses to Questions

1. Is the manuscript technically sound, and do the data support the conclusions?

Reviewer #1: Yes

Reviewer #2: Partly

2. Has the statistical analysis been performed appropriately and rigorously?

Reviewer #1: Yes

Reviewer #2: I Don't Know

3. Have the authors made all data underlying the findings in their manuscript fully available?

Reviewer #1: Yes

Reviewer #2: No

4. Is the manuscript presented in an intelligible fashion and written in standard English?

Reviewer #1: Yes

Reviewer #2: Yes

5. Review Comments to the Author

Comments to the Author

Reviewer #1: This manuscript explored the use of fasting glucose and glucose tolerance test for diabetes diagnosis in mice. Here are some questions:

1. The abstract needs more specific results, e.g., by which week MetS develops and based on what parameters, body weight increase by what percentage can be used as a predictor of hyperglycemia.

-> Thank you for the comment. We added the detailed information on metabolic syndrome diagnostic parameters as well as “cut-off” value of body weight gain to the Abstract section, line 27 – 30, and line 33 – 34. We now write:

Line 27 – 30: “After consuming high-fat diet for 4 weeks, mice developed metabolic syndrome, including obesity, significant increase of fasting plasma cholesterol level, elevation of both C-peptide and fasting blood glucose levels.” 

Line 33 – 34: “Besides, among metabolic measurements, accumulating body weight gain > 16.23 g for 12 weeks could be utilized as a potential parameter to predict type 2 diabetes development in C57BL/6J mice.”

2. Results. Line 117-118. By which week the weight gain was different between high-fat and ND group? This is probably more important than comparison to the baseline given that there is healthy weight gain even when the mice are on an ND.

-> Thank you for the comment. The body weight gain of HFD group was significantly higher than ND group by the second week as shown in Fig 1A. We also agree with the reviewer that during 20 weeks, mice from both HFD and ND group gained weight. While ND group showed a healthy weight gain, the HFD group gained weight more rapidly. And the interpretation was revised in Result section, line 124 – 126. We now write:

“Over a twenty-week period, mice gradually gained weight regardless of the diet type (Fig 1A). As expected, mice fed with HFD became significantly heavier than those fed with ND as early as the second week (p = 0.02) (Fig 1A).”

3. Line 126. Was there any difference in calorie intake?

-> Thank you for the comment. The HFD showed significant higher calorie intake. To demonstrate this, we replaced food intake data in the previous Fig 1E from calculating as gram/mouse/day into kcal/mouse/day in a new Fig 1E, based on the different energy amount per gram of each diet to demonstrate the difference in calorie intake (2.97 kcal/g for ND, and 4.7 kcal/g for HFD). The information on kcal/g of each type of diet was added in the Method section, line 80 – 82. The interpretation regarding energy consumption was revised accordingly to the new Fig 1E at the Result section, line 140 – 141. We now write: 

Line 140 – 141: “Weekly food and water monitoring showed more energy consumption rate and less water intake in the HFD group (Fig 1E-G).” 

New Fig 1 E:

(Please see the "Response to reviewer" file or Fig1 file) 

4. Line 127. What are the criteria for MetS diagnosis? It seems that at week 4 only weight and cholesterol were significantly different between HF and ND, doesn’t MetS diagnosis also need a blood pressure or blood glucose measurement? A relevant marker C-peptide was not different until week 8.

-> Thank you for the comment. MetS was defined as obesity, dyslipidemia, hyperinsulinemia, and elevated fasting blood glucose. We had shown the elevation of fasting blood glucose level as well as glucose intolerance test at 4 weeks in Fig 2A and Fig 2B, respectively. However, in the Results section, “Metabolic syndrome development in 45% HFD-fed C57BL/6J mice” part, blood glucose measurement was not mentioned. To demonstrate more clearly the MetS diagnosis in C57BL/6J, we added the interpretation on fasting blood glucose level together with S1 Table in line 138 – 140. We now write:

“The mean value of F-BGL of HFD group also elevated higher than that of the ND group at the fourth week, yet the level still sustained below 200 mg/dL (S1 Table).”

S1 Table: F-BGL changes during 20 weeks

(Please see the "Response to reviewer" file or Supporting Information S1 Table file) 

Some typos need to be corrected

Line 24. “are” is missing after “mouse model”; line 42. “of” should be deleted

-> Thank you for the comment. We revised typos accordingly. “are” was added in line 25; “of” in line 44 was removed.

Reviewer #2: This study aims to establish criteria with which to distinguish progressive stages in the development of diet-induced Type 2 diabetes (T2D) in mouse studies, with which investigators might more accurately focus studies on prevention and treatment. They do this by examining fasting blood glucose levels, oral glucose tolerance test performance, and body weight gain in male C57BL/6J mice, a popular model for studying diet-induced obesity, over a course of 20 weeks on high-fat diet. The results are potentially useful in the study of the development of T2D should these stages prove to be as distinct as purported. However, the manuscript as written has several deficiencies that should be resolved in order to be thoroughly convincing.

Major

1. The main goal of the study is to distinguish metabolic syndrome from pre-diabetes and eventually from onset of type 2 diabetes. The authors appear to propose that metabolic syndrome (characterized by obesity, dyslipidemia, and slight increase in fasting plasma C-peptide) differs from pre-diabetes by the lack of both elevated fasting blood glucose level (F-BGL) or significant elevation of 2-hour oral glucose tolerance (OGTT) values above “normal” levels, while pre-diabetes is characterized by elevated F-BGL intermediate but significantly elevated 2-hour OGTT, and diabetes with elevated F-BGL and even higher OGTT levels. However, the purported purpose of this study is to propose criteria for classifying type 2 diabetes progressive stages, yet operational definitions of these stages are never defined. For example, at what point in another group’s study would one consider the mice pre-diabetic versus diabetic?) It would be helpful to explicitly and clearly state these criteria if indeed this is the main assertion.

-> Thank you for the comment. We revised the type 2 diabetes progressive stages in comparison with previous studies at the Discussion section, line 235 – 238 (for prediabetes), and line 243 – 245 (for the type 2 diabetes). In brief, addition to other metabolic parameters, our study suggested a more detailed range of 2-hour OGTT from 250 mg/dL to 300 mg/dL for prediabetes, and over 300 mg/dL for type 2 diabetes diagnosis in HFD-fed C57BL/6J male mice. We now write: 

“In rodents, several parameters were used to define prediabetes, including glucose intolerance, hyperinsulinemia, hyperlipidemia, mild hyperglycemia and increased 2-hour glucose tolerant test, yet the detailed criteria for diagnosis is still lacking (9). In this study, 45% HFD fed C57BL/6J mice from 8 to 20 weeks showed two-stage elevation of 2-hour-OGTT with cut-point of approximately 300 mg/dL. Thus, our data proposes that 45% HFD induced pre-diabetes from the eighth week with 2-hour-OGTT range from 250 to 300 mg/dL before turning to the onset of T2D from the sixteenth week with 2-hour-OGTT over 300 mg/dL in C57BL/6J mice. Indeed, 2-hour OGTT in mice was suggested to reach at least 300 mg/dL to be equivalent to the T2D diagnostic criteria of 200 mg/dL in human that might be due to the higher level of basal blood glucose level in mice (14).”

2. This study only addresses development of diabetes in male mice in this model, which differs in several relevant regards from female responses in several of the measures analyzed (e.g., weight gain in females is much less dramatic during development of diabetes). Conclusions of this study should be clearly explicitly confined to male mice.

-> Thank you for the comment. We agree with the suggestion from the reviewer. We revised the conclusion of the study that the result is specific in “C57BL/6J male mice” at the line 265. We now write:

“In summary, our study illustrated an overview of the early progressive stage from MetS and pre-diabetes to onset T2D in C57BL/6J male mice by feeding with 45% HFD, which might support future investigations in the field of T2D prevention and treatment at the early stage of the disease.”

3. Please clarify which factors were included in the two-way ANOVAs employed in the analyses, as well as associated F-values, degrees of freedom and “n” for the tests in question.

-> Thank you for the comment. We added detailed information on which factors were used for two-way ANOVAs analysis in the Method section, Data analysis part, line 112 – 115. We now write:

“The comparisons between HFD and ND groups on the change over time of body weight, cholesterol, triglyceride, C-peptide, energy and water intake, F-BGL, 2-hour-OGTT were assessed using two-way analysis of variance (ANOVA), followed by Holm-Šídák multiple comparisons test for each time point.”

We also add the total amount of mice as well as grouping information with “n” for subsequence experiment in Methods section, line 83 - 85. We now write:

“A total of 64 mice were divided as 4 mice per cage, 8 cages per group. 1 mouse from ND group was excluded due to weight loss and was excluded from all analysis.”

Regarding associated F-values, Degree of freedom and “n” for each tests in question:

a. Body weight (Fig 1A): 

Factors: “Body weight” and “Time”

Subjects: ND (n = 31), HFD (n = 32)

F(1, 61) = 164.5, p < 0.001

Missing values: 0

b. Cholesterol (Fig 1B):

Factors: “Cholesterol” and “Time”

Subjects: ND (n = 8), HFD (n = 8)

F(1, 14) = 133.2, p < 0.001

Missing values: 0

c. Triglyceride (Fig 1C):

Factors: “Triglyceride” and “Time”

Subjects: ND (n = 8), HFD (n = 8)

F(1, 14) = 50.71, p < 0.001

Missing values: 0

d. C-peptide (Fig 1D):

Factors: “C-peptide” and “Time”

Subjects: ND (n = 8), HFD (n = 8)

F(1, 14) = 62.31, p < 0.001

Missing values: 3 (1 value from ND group week 20; 2 values from HFD group week 20)

e. Calorie intake (Fig 1E): 

Factors: “Calorie intake” and “Time”

Subjects: ND (n = 8), HFD (n = 8) 

F(1, 14) = 73.34, p < 0.001

Missing values: 0

f. Water intake (Fig 1F):

Factors: “water intake” and “Time”

Subjects: ND (n = 8), HFD (n = 8) 

F(1, 14) = 103.2, p < 0.001

Missing values: 0

g. F-BGL (Fig 2A):

Factors: “F-BGL” and “Time”

Subjects: ND (n = 31), HFD (n = 32) 

F(1, 61) = 152.6, p < 0.001

Missing values: 0

h. 2h-OGTT (Fig 2G):

Factors: “2h-OGTT” and “Time”

Subjects: ND (n = 8), HFD (n = 8) 

F(1, 14) = 71.38, p < 0.001

Missing values: 0

4. In the description of “Deterioration of glycemic control during 20 weeks consuming 45% HFD”, authors refer to “area under the curve” of the OGTT (line 139), yet the data presented in the accompanying Fig. 2 are not AUC data (but would be a relevant analysis). Moreover, if AUC data were analyzed, the method of calculating AUC should be described in the Methods section.

-> Thank you for the comment. We added the AUC data for each OGTT result in Fig 2B-D in the new S2 Fig A-E as below (next page). We also mentioned how the AUC was calculated in the Method section as indicated in line 115 – 117. We now write: 

“For oral glucose tolerance test, area under the curve (AUC) was calculated by trapezoid rule, and the comparison was done by two-tailed student's t-test.”

S2 Fig. (Please see the "response to reviewer" file or Supporting Information S2 Fig file)

5. The authors’ make little mention of the methods of the correlational multi-variable analysis, the results of which suddenly appear in the manuscript (line 156). Similarly, there is no description of the receiver operating characteristic curve (ROC) analysis (line 163). These descriptions are essential for appropriate evaluation of these results.

-> Thank you for the comment. We agree with the reviewer that the description for these data in the result section was too brief, although we also mentioned in the legend of figure 3. To make it clearer, we included the explanation on why and how we conducted multi-variable analysis in line 187 -199, as well as ROC analysis in line 200 – 205 in Results section. We now write:

For multi-variable analysis: 

“Our above data demonstrated the worsening of metabolic parameters coincident with failure from mild to moderate glycemic control during a period of 20 weeks in C57BL/6J male mice fed with 45% HFD. We questioned which metabolic factor tested above, including weight gain, cholesterol, and triglyceride could predict the loss of glycemic control in this HFD mouse model. First, the association of those factors with F-BGL during 20 weeks was examined using mice with a full diagnostic panel (weight gain, cholesterol, triglyceride, and F-BGL) at week 4 (n = 16), week 12 (n = 29), week 16 (n = 16), and week 20 (n = 16). A mouse from each time point was considered as an independent subject, thus a total of n = 77 subject was input for multi-variables analysis. Matrix correlation result showed a positive correlation among body weight, fasting plasma cholesterol and F-BGL (Fig 3A). This data confirmed HFD-fed C57BL/6J male mouse model reflex the human disease, in which individuals having MetS and/ or pre-diabetes are at high risk of T2D development.”

For ROC analysis:

“Next, because weight gain had a stronger positive correlation with F-BGL compared to fasting plasma cholesterol, we identified which amount of body weight gain could predict the decline in glycemic control of C57BL/6J mice. Since mice developed mild glycemic control failure from week 8 with 2h-OGTT over 250 mg/dL (Fig 2G), at week 8, 12, 16 and 20, mice were grouped depending on their F-BGL either less than 200 mg/dL (as normal glucose level) or equal to and more than 200 mg/dL (abnormal glucose level) for receiver operating characteristic curve (ROC) analysis.”

6. The “cut-offs” for body weight gain at which body weight predicted loss of glycemic control (line 166) are stated in absolute values (e.g., 37.6g). Given that the weights at which the stages of diabetes development were unique to this specific cohort of mice given HFD at a specific age, would it not be more useful to other groups to determine relative weight gain cut-offs (i.e., relative to starting weights? Or weight gain relative to that in control mice?) that would be more extrapolatable to studies in which there is some variability from this study, such as in the age at which HFD is begun, in the starting weights of the mice, the rate of growth in the different groups in the study, or other factors? (For example, if one were to perform a study where mice received HFD at 5 weeks of age, would 37.6g be the body weight at which this tipping point occurs?)

-> Thank you for raising this point. We totally agreed with the reviewer’s opinion on the usefulness of relative weight gain for other studies if the HFD protocol would be slightly different in terms of initial body weight. Therefore, we re-calculated both multi-variables analysis and ROC analysis to identify the “cut-off” of relative weight gain to their initial body weight at week 0 (six-week of age). Interestingly, the accumulating of weight gain is likely to predict the glycemic control failure right at week 12 (earlier than the absolute body weight). We changed the Fig 3 with relative weight gain data instead of absolute body weight. We also revised the Result section, line 206 – 214, and the Discussion section, line 258 according to the new Fig 3. We now write:

Line 206 – 214: 

“Area under the ROC curve revealed that weight gain at week 12 is likely to predict a mouse's failure to control its F-BGL (AUC = 0.9373), with suggestion of “cut-off” for weight gain > 16.23 g (85.71% of sensitivity and 90.24% of specificity, likelihood ratio 8.786) (Fig 3B-E). To validate this "cut-off" value, we re-analyzed the 2-hour-OGTT in the HFD group at week 16 and 20. Interestingly, mice gained less than 16.23 g at week 12 had 2-hour-OGTT at week 16 and 20 below 300 mg/dL (Fig 3F). These data suggested that although mice fed with 45% HFD developed MetS together with impairment of glycemic control, those that gained weight over 16.23 g after 12 weeks were likely to develop hyperglycemia.”

Line 258:

 “In addition, our data suggested accumulating weight gain over 16.23 g after 12 weeks fed with 45% HFD as the possible predictor of T2D development in C57BL/6J mice.” 

New Fig 3:

(Please see the "Response to reviewer" file or Fig3 file) 

7. It is unclear from Ref. 14 how “six-hour fasting from 8 am – 2 pm” resembles human physiological conditions (line 190). Is it the duration of the fast, or the timing of the fast that is deemed relevant here? Does the fact that mice are nocturnal and normally sleeping (not eating) at 8 am not put this method at odds with human conditions of the OGTT?

-> Thank you for the comment. “six-hour fasting from 8 am – 2 pm” resembles human physiological conditions in terms of fasting during circadian sleeping time. And the previous ref.14 was cited to explain on the physiological mechanism of glucose absorption through oral gavage. We agree that the explanation was not clear enough. We revised the explanation in the Discussion section, line 245 – 253, the ref. 14 now becomes ref.16, and we cited a new ref.15 for explanation of six hours during 8 am – 2 pm. We now write:

“Of note, mice are nocturnal feeders with a majority of food intake occurring during the dark cycle, and overnight fasting was shown to enhance insulin sensitivity and glucose uptake by muscle (15). Additionally, glucose administration through oral gavage is absorbed by gastrointestinal tract, leading to gut incretin hormones-stimulated insulin secretion (16). In this study, the glucose tolerance test was conducted after a six-hour fasting from 8 am to 2 pm through oral gavage to resemble human physiological conditions, the range of the 2-hour glucose tolerance test obtained either from a longer fasting duration or by intraperitoneal injection method might need to be evaluated separately.”

Ref. (15) Heijboer AC, Donga E, Voshol PJ, Dang ZC, Havekes LM, Romijn JA, et al. Sixteen hours of fasting differentially affects hepatic and muscle insulin sensitivity in mice. J Lipid Res. 2005;46(3):582-8.

(Previous ref. 14) Ref. (16) Drucker DJ. Incretin action in the pancreas: potential promise, possible perils, and pathological pitfalls. Diabetes. 2013;62(10):3316-23.

8. The authors claim that all data is contained within the manuscript, yet only summary statistics are included.

-> Thank you for the comment. We added the section “Supporting Information” to the revised manuscript for the detailed data information, which were shown as summary statistics in the previous submitted manuscript. In brief, we added table 1 for relative body weight gain (see answer to question 11, reviewer #2 below), and S1 Table for F-BGL during 20 weeks (see answer to question 4, reviewer #1 above) 

Minor issues:

9. Fasting blood glucose levels were not “tested… followed by six-hour fasting.” (line 83) Presumably the levels were tested following six-hour fasting?

-> Thank you for the comment. We corrected as below: 

“Blood glucose levels were tested after six-hour fasting by One Touch Ultra glucometer (Lifescan, Inc., Chesterbrook, PA, USA).” (Line 86 – 87) 

10. It is unclear what the authors mean by “cutting off the tail vein” (line 92). Does this mean snipping off the end of the tail, rather than via a puncture method? Please define this more clearly.

-> Thank you for the comment. We corrected as below: 

“Blood glucose levels were measured… by snipping off the end of the tail” (Line 96)

11. Authors claim that “mice fed with HFD gradually gained weight” (line 116). From the data, it appears that both groups of mice gained weight, although HFD mice gained more.

-> Thank you for the comment. We agreed with the reviewer that both mice from ND and HFD gained weight over a period of 20 weeks. However, the weight gain rate in HFD group was more rapid than the ND group. To make this comparison clear, we added the relative weight gain (table 1 and S1 Fig). The interpretation was revised in the Results section, line 126 – 131. We now write: 

 “When comparing the relative weight gain to their initial body weight at six-week of age, HFD group showed an increase rate more rapid than ND group (Table 1). The mean value of the relative weight gain of the HFD group reached to the mean value + 3 standard deviations of the ND group at the fourth week (S1 Fig). This result suggested that C57BL/6J male mice became obesity after four weeks of consuming a 45% HFD.”

Table 1: Relative weight gain to the initial body weight during 20 weeks

 (Please see the "Response to reviewer" file or Table 1 file)

S1 Fig: The relative weight gain over 20 weeks

 (Please see the "Response to reviewer" file or Supporting Information S1 Fig file)

6. PLOS authors have the option to publish the peer review history of their article (what does this mean?). If published, this will include your full peer review and any attached files.

Do you want your identity to be public for this peer review? For information about this choice, including consent withdrawal, please see our Privacy Policy.

Reviewer #1: No

Reviewer #2: Yes: E. Todd Weber

We look forward to hearing from you in due time regarding our submission and to respond to any further questions and comments you may have.

Sincerely,

Chung-Gyu Park, MD, PhD

Professor, Seoul National University College of Medicine

103 Daehak-ro Jongno-gu, Seoul 03080, Korea

Tel. +82-2-740-8007, 8308 Fax +82-2-743-0881

E-mail: chgpark@snu.ac.kr, chgpark@gmail.com

---

## [Decision Letter · Decision Letter 1]

17 Oct 2023

PONE-D-23-24628R1Determination of progressive stages of type 2 diabetes in a 45% high-fat diet-fed C57BL/6J mouse model is achieved by utilizing both fasting blood glucose levels and a 2-hour oral glucose tolerance testPLOS ONE

Dear Dr. Park,

Thank you for submitting your manuscript to PLOS ONE. After careful consideration, we feel that it has merit but does not fully meet PLOS ONE’s publication criteria as it currently stands. Therefore, we invite you to submit a revised version of the manuscript that addresses the points raised during the review process.

We look forward to receiving your revised manuscript.

Kind regards,

Peng Gao, Ph.D.

Academic Editor

PLOS ONE

Journal Requirements:

Additional Editor Comments:

There are two remaining issues.

1) The authors have included their F-values and degrees of freedom in their response to reviewers, but have not embedded them within the manuscript.

2) The availability of raw data should be declared in the methods section.

Reviewers' comments:

Reviewer's Responses to Questions

**Comments to the Author**

1. If the authors have adequately addressed your comments raised in a previous round of review and you feel that this manuscript is now acceptable for publication, you may indicate that here to bypass the “Comments to the Author” section, enter your conflict of interest statement in the “Confidential to Editor” section, and submit your "Accept" recommendation.

Reviewer #1: All comments have been addressed

Reviewer #2: (No Response)

2. Is the manuscript technically sound, and do the data support the conclusions?

Reviewer #1: Yes

Reviewer #2: Yes

3. Has the statistical analysis been performed appropriately and rigorously? 

Reviewer #1: Yes

Reviewer #2: Yes

4. Have the authors made all data underlying the findings in their manuscript fully available?

Reviewer #1: Yes

Reviewer #2: No

5. Is the manuscript presented in an intelligible fashion and written in standard English?

Reviewer #1: Yes

Reviewer #2: Yes

6. Review Comments to the Author

Reviewer #1: This study sought to determination of progressive stages of type 2 diabetes using a mouse model. Thank you for addressing my questions. I do not have further comments.

Reviewer #2: (No Response)

7. PLOS authors have the option to publish the peer review history of their article (what does this mean?). If published, this will include your full peer review and any attached files.

Reviewer #1: No

Reviewer #2: No

---

## [Author Response · Author response to Decision Letter 1]

18 Oct 2023

October 18th, 2023

Dear Dr. Peng Gao

PLOS ONE Academic Editor,

We really appreciate to re-submit our manuscript entitled "Determination of progressive stages of type 2 diabetes in a 45% high-fat diet-fed C57BL/6J mouse model is achieved by utilizing both fasting blood glucose levels and a 2-hour oral glucose tolerance test" [PONE-D-23-24628R]. We are grateful to the reviewers and editors for their helpful and supportive comments. We have addressed the issue raised by the editor in the response letter and the revised manuscript with the track changes. Here is a point-by-point response to the editor’s comments.

Journal Requirements:

> Thank you for the comment. We have checked all the references that were cited in this manuscript. As of now, none of them is a retracted paper.

Additional Editor Comments:

There are two remaining issues.

1) The authors have included their F-values and degrees of freedom in their response to reviewers, but have not embedded them within the manuscript.

> Thank you for the comment. We added F-values and degrees of freedom in the S1 File at the Supporting information section in the revised manuscript, line 331-332. 

2) The availability of raw data should be declared in the methods section.

> Thank you for the comment. We removed the “Data availability” in the methods section, lines 118 – 120. Raw data was provided in the S1 File in the Supporting information section in the revised manuscript, lines 331-332.

Reviewers' comments:

Reviewer's Responses to Questions

Comments to the Author

1. If the authors have adequately addressed your comments raised in a previous round of review and you feel that this manuscript is now acceptable for publication, you may indicate that here to bypass the “Comments to the Author” section, enter your conflict of interest statement in the “Confidential to Editor” section, and submit your "Accept" recommendation.

Reviewer #1: All comments have been addressed

> Thank you for your comment.

Reviewer #2: (No Response)

2. Is the manuscript technically sound, and do the data support the conclusions?

Reviewer #1: Yes

Reviewer #2: Yes

3. Has the statistical analysis been performed appropriately and rigorously?

Reviewer #1: Yes

Reviewer #2: Yes

4. Have the authors made all data underlying the findings in their manuscript fully available?

Reviewer #1: Yes

Reviewer #2: No

5. Is the manuscript presented in an intelligible fashion and written in standard English?

Reviewer #1: Yes

Reviewer #2: Yes

6. Review Comments to the Author

Reviewer #1: This study sought to determination of progressive stages of type 2 diabetes using a mouse model. Thank you for addressing my questions. I do not have further comments.

> Thank you for your comment.

Reviewer #2: (No Response)

7. PLOS authors have the option to publish the peer review history of their article (what does this mean?). If published, this will include your full peer review and any attached files.

Do you want your identity to be public for this peer review? For information about this choice, including consent withdrawal, please see our Privacy Policy.

Reviewer #1: No

Reviewer #2: No

We hope that you will find the revised manuscript satisfactory for publication. We look forward to hearing from you in due time regarding our submission and to respond to any further questions and comments you may have.

Sincerely,

Chung-Gyu Park, MD, PhD

Professor, Seoul National University College of Medicine

103 Daehak-ro Jongno-gu, Seoul 03080, Korea

Tel. +82-2-740-8007, 8308 Fax +82-2-743-0881

E-mail: chgpark@snu.ac.kr, chgpark@gmail.com

---

## [Editor Report · Decision Letter 2]

23 Oct 2023

Determination of progressive stages of type 2 diabetes in a 45% high-fat diet-fed C57BL/6J mouse model is achieved by utilizing both fasting blood glucose levels and a 2-hour oral glucose tolerance test

PONE-D-23-24628R2

Dear Dr. Park,

We’re pleased to inform you that your manuscript has been judged scientifically suitable for publication and will be formally accepted for publication once it meets all outstanding technical requirements.

Kind regards,

Peng Gao, Ph.D.

Academic Editor

PLOS ONE
---

## [Editor Report · Acceptance letter]

25 Oct 2023

PONE-D-23-24628R2 

Determination of progressive stages of type 2 diabetes in a 45% high-fat diet-fed C57BL/6J mouse model is achieved by utilizing both fasting blood glucose levels and a 2-hour oral glucose tolerance test  

Dear Dr. Park:

I'm pleased to inform you that your manuscript has been deemed suitable for publication in PLOS ONE. Congratulations! Your manuscript is now with our production department. 

Kind regards, 

on behalf of

Professor Peng Gao 

Academic Editor

PLOS ONE